# Endoplasmic reticulum-derived bodies enable a single-cell chemical defense in Brassicaceae plants

Kenji Yamada [1,2,3]*, Shino Goto-Yamada[1,2,3], Akiko Nakazaki[3], Tadashi Kunieda[3,4,5], Keiko Kuwata[6], Atsushi J. Nagano [7], Mikio Nishimura[2,4]* & Ikuko Hara-Nishimura [3,4]*

Brassicaceae plants have a dual-cell type of chemical defense against herbivory. Here, we show a novel single-cell defense involving endoplasmic reticulum (ER)-derived organelles (ER bodies) and the vacuoles. We identify various glucosinolates as endogenous substrates of the ER-body β-glucosidases BGLU23 and BGLU21. Woodlice strongly prefer to eat seedlings of *bglu23 bglu21* or a glucosinolate-deficient mutant over wild-type seedlings, confirming that the β-glucosidases have a role in chemical defense: production of toxic compounds upon organellar damage. Deficiency of the Brassicaceae-specific protein NAI2 prevents ER-body formation, which results in a loss of BGLU23 and a loss of resistance to woodlice. Hence, NAI2 that interacts with BGLU23 is essential for sequestering BGLU23 in ER bodies and preventing its degradation. Artificial expression of NAI2 and BGLU23 in non-Brassicaceae plants results in the formation of ER bodies, indicating that acquisition of NAI2 by Brassicaceae plants is a key step in developing their single-cell defense system.

[1] Malopolska Centre of Biotechnology, Jagiellonian University, 30-387 Krakow, Poland. [2] Department of Cell Biology, National Institute for Basic Biology, Okazaki 444-8585, Japan. [3] Graduate School of Science, Kyoto University, Kyoto 606-8502, Japan. [4] Faculty of Science and Engineering, Konan University, Kobe 658-8501, Japan. [5] Graduate School of Science and Technology, Nara Institute of Science and Technology, Ikoma 630-0192, Japan. [6] Institute of Transformative Bio-Molecules, Nagoya University, Nagoya 464-8601, Japan. [7] Faculty of Agriculture, Ryukoku University, Otsu, Shiga 520-2194, Japan. *email: kenji.yamada@uj.edu.pl; mikosome@nibb.ac.jp; ihnishi@gr.bot.kyoto-u.ac.jp

Plants have evolved many defense strategies against herbivores and pathogens, including the production and release of toxic compounds. Brassicaceae plants have a chemical herbivory defense system (called mustard-oil bomb) involving thioglucosidases (also called myrosinases) and their substrates glucosinolates. In *Arabidopsis thaliana* (a Brassicaceae plant), myrosinases (TGG1 and TGG2) accumulate in myrosin cells along the vasculature of mature leaves[1,2], while glucosinolates accumulate in other cells called S cells[3]. When herbivores damage tissues, myrosinases gain access to glucosinolates and hydrolyze them to produce the toxic compounds isothiocyanates[4,5]. Thus, the myrosinase–glucosinolate system is a dual-cell type of chemical defense.

In contrast to the abundance of TGG1 and TGG2 in mature leaves, neither enzyme is detectable in *A. thaliana* seedlings[6]. Instead, seedlings have large amounts of another type of β-glucosidase (BGLU23, also known as PYK10) that is a major component of the endoplasmic reticulum (ER)-derived organelles called ER bodies[7–10]. An in vitro analysis showed that BGLU23 has β-glucosidase activity toward *O*- and *S*-linked glucosides[11,12]. Upon tissue damage, BGLU23 forms complexes with a cytosolic lectin (PYK10-binding protein 1), resulting in enhancement of β-glucosidase activity[13,14]. However, the endogenous substrates of BGLU23 are unknown.

ER bodies are ~10-μm-long, spindle-shaped structures that occur in Brassicaceae plants[9,12,15,16]. They are distributed throughout the epidermal cells of cotyledons and hypocotyls, but subsequently disappear with seedling growth[7], while they constitutively develop in the epidermal cells of roots[8]. On the other hand, mature leaves have no ER bodies, but they are induced by wounding or treatment with the wound hormone jasmonate[7]. The findings that ER bodies are distributed in epidermal cells (which are easily attacked by herbivores and pathogens[9,10] and that they are induced by wounding lead us to propose that ER bodies function in the defense against pests.

*A. thaliana*'s closest BGLU23 homolog (BGLU21) is also localized to ER bodies. Reduction of the levels of BGLU23 and BGLU21 causes ER bodies to become elongated, suggesting that ER-body morphology is modulated by the contents[17]. A prominent feature of BGLU23 and BGLU21 is that both have an ER-retention signal (Lys-Asp-Glu-Leu (KDEL)) at the C-terminus[17], which occurs in ER-resident proteins in eukaryotic cells of plants, yeast, and animals[18–21]. However, in plants, ER-retention signals are not always enough to retain proteins in the ER[22,23]. The BGLU proteins are abundantly synthesized on the ER after seed germination[8]. The de novo synthesized proteins are localized to the ER bodies by an unknown mechanism.

In this study, we describe a novel single-cell type of chemical defense, in which tissue damage brings ER-body β-glucosidases into contact with substrates stored in the vacuoles, identify various glucosinolate species as endogenous substrates of BGLU23 and BGLU21 and demonstrate how the Brassicaceae-specific proteins NAI2 and BGLU23 can induce ER-body formation in non-Brassicacea plants.

## Results

**Native substrate glucosinolates of ER-body β-glucosidases**. To elucidate the function of ER bodies, we first focused on their major component, BGLU23, in *A. thaliana* seedlings. We determined the native substrates of the ER-body β-glucosidases, by comparing the metabolomes of the wild type and the β-glucosidase-deficient mutant *bglu23 bglu21*, which lacks BGLU23 and its homolog BGLU21[17]. BGLU21 is a less abundant β-glucosidase of ER bodies[13]. Metabolites in the seedling homogenates were analyzed before and after incubation at 26 °C for 30 min to allow any β-glucosidases present to react with their substrates. We identified a total of 1406 metabolites, for each of which we obtained four MS signal intensities: before and after incubation of the wild-type homogenate ($I_{WT,0}$ and $I_{WT,30}$, respectively) and before and after incubation of the *bglu23 bglu21* homogenate ($I_{bglu,0}$ and $I_{bglu,30}$, respectively). Among the 1406 metabolites, 76 had intensity profiles, in which $I_{WT,0} > I_{WT,30}$ and $I_{bglu,30} > I_{WT,30}$ (Supplementary Data 1), indicating that their levels decreased during incubation in a β-glucosidase-dependent manner. Of these 76 metabolites, 13 were identified as glucosinolates, including eight aliphatics, four aromatics, and one indole (Table 1 and Supplementary Table 1). Most of the 13 glucosinolates disappeared after 30 min incubation of the wild-type homogenate, but not after incubation of the

**Table 1 Changes in mass spectrometry signal intensities of 13 glucosinolates in the wild type and *bglu23 bglu21* seedling homogenates before and after 30 min at 26 °C.**

| Glucosinolates | Wild type | | *bglu23 bglu21* | |
|---|---|---|---|---|
| | MS signal (log2) | | MS signal (log2) | |
| | Before incubation ($I_{WT,0}$) | After incubation ($I_{WT,30}$) | Before incubation ($I_{bglu,30}$) | After incubation ($I_{bglu,30}$) |
| Aliphatic glucosinolates | | | | |
| 4MTB | 23.34 ± 0.21 | 13.88 ± 0.38* | 23.05 ± 0.08 | 22.22 ± 0.27 |
| 5MTP | 20.39 ± 0.28 | n.d. | 20.89 ± 0.07 | 20.07 ± 0.28 |
| 6MTH | 18.95 ± 0.34 | n.d. | 20.87 ± 0.07 | 19.73 ± 0.40 |
| 7MTH | 21.21 ± 0.19 | n.d. | 22.85 ± 0.03 | 21.60 ± 0.43 |
| 8MTO | 22.54 ± 0.32 | 4.61 ± 4.61* | 23.91 ± 0.03 | 22.55 ± 0.49 |
| 4MSOB | 15.70 ± 0.58 | n.d. | 16.26 ± 0.18 | 14.26 ± 0.57 |
| 5MSOP | 14.47 ± 0.29 | n.d. | 14.03 ± 0.37 | 13.99 ± 0.29 |
| 7MSOH | 16.45 ± 0.34 | n.d. | 17.72 ± 0.48 | 17.69 ± 0.26 |
| Aromatic glucosinolates | | | | |
| 2PE | 16.59 ± 0.33 | n.d. | 16.75 ± 0.08 | 15.39 ± 0.43 |
| 3BzOP | 21.47 ± 0.19 | n.d. | 21.25 ± 0.06 | 19.92 ± 0.41 |
| 4BzOB | 22.58 ± 0.15 | n.d. | 24.18 ± 0.09 | 22.64 ± 0.50 |
| 5BzOP | 15.82 ± 0.21 | n.d. | 17.84 ± 0.06 | 16.86 ± 0.36 |
| Indole glucosinolates | | | | |
| I3M | 19.83 ± 0.21 | n.d. | 20.84 ± 0.08 | 20.39 ± 0.22 |

Standard error of three independent experiments are shown. See Supplementary Table 1 for abbreviations of glucosinolates. Before and after signals that are significantly different ($p < 0.05$; two-sided Student's *t* test) are marked with asterisks
n.d. not detectable

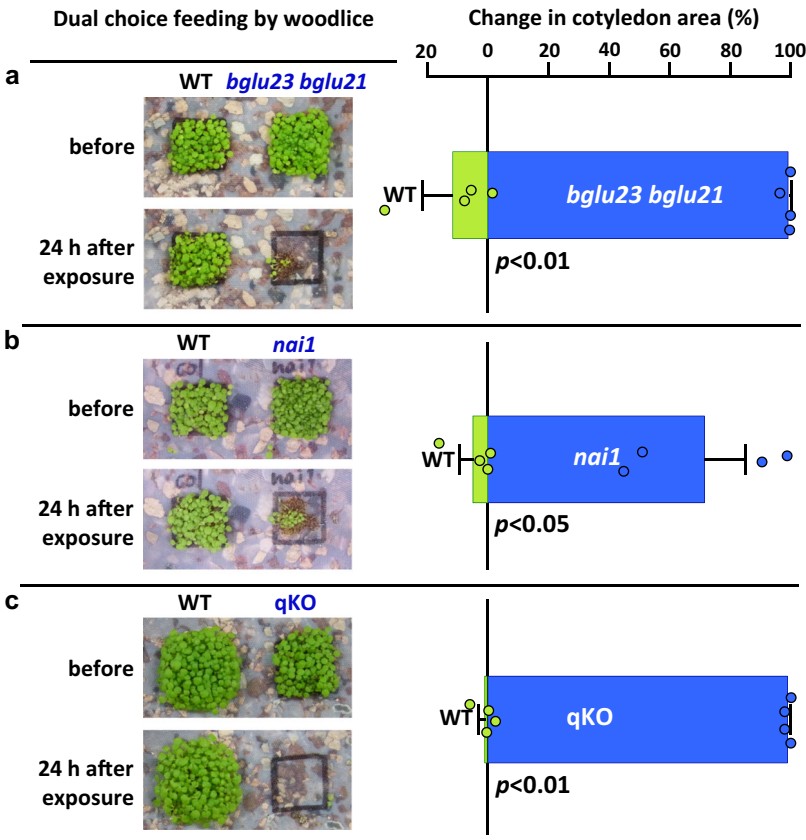

**Fig. 1 Effects of ER-body-β-glucosidases and glucosinolates on woodlouse feeding on *A. thaliana* seedlings.** The photos compare changes in the cotyledon area of seedlings before and 24 h after exposure to fasted woodlice (*A. vulgare*) in wild type (WT) and three mutants (*bglu23 bglu21*, *nai1*, and qKO). The reduction of cotyledon area by feeding is shown. Error bars indicate standard error of four independent experiments. Significance values were calculated by two-sided Student's *t* test. See Supplementary Data 3 for source data. **a** ER-body β-glucosidase-deficient mutant *bglu23 bglu21*. **b** ER-body defective mutant *nai1*. **c** Glucosinolates-deficient quadruple mutant *myb28 myb29 cyp79b2 cyp79b3* (qKO).

*bglu23 bglu21* homogenate (Table 1). These glucosinolates are reported to be major glucosinolates in *A. thaliana* seeds[24], indicating that BGLU23 and BGLU21 function as major glucosinolate-converting β-glucosidases of seedlings.

**ER-body β-glucosidases and glucosinolates against predators.** Glucosinolates are components of a dual-cell chemical defense system in mature leaves of Brassicaceae plants, in which myrosinases react with glucosinolates to form toxic compounds isothiocyanates that deter herbivory[4,25,26]. Myrosinases are β-glucosidases that belong to a subfamily different from the subfamily containing BGLU23 and BGLU21[1,12]. To determine whether BGLU23 and BGLU21 have a role in seedling defense against animals, we used adult woodlice (*Armadillidium vulgare*) as a model herbivore. In a dual-choice feeding assay, they were offered wild-type and *bglu23 bglu21* seedlings as food. The woodlice, even when fasted, hardly touched the wild-type seedlings, but ate virtually all the *bglu23 bglu21* seedlings in 24 h (Fig. 1a). The woodlice also fed on an *A. thaliana* mutant (*nai1*) that lacks the transcription factor NAI1 regulating the expression of BGLU23[27] (Fig. 1b). We next gave woodlice a choice between seedlings of the wild type and the quadruple mutant *myb28 myb29 cyp79b2 cyp79b3* (qKO), which is defective in synthesis of the major glucosinolates[28]. Fasted woodlice fed almost exclusively on qKO (Fig. 1c). These results clearly demonstrate that ER-body β-glucosidases and glucosinolates can defend seedlings against woodlice. Hence, woodlice avoid the toxic compounds isothiocyanates that are produced from glucosinolates by the β-glucosidases BGLU23 and BGLU21.

**NAI2 and BGLU23 regulate the ER-body formation.** ER bodies are unique to Brassicaceae plants[9]. Unexpectedly, however, we found that artificial expression of the Brassicaceae-specific proteins BGLU23 and NAI2 induced the formation of ER bodies in non-Brassicaceae plants including a monocot (onion) and a dicot (tobacco). NAI2 is an ER-body component that has ten repeats of ~40-amino acid sequence containing an acidic motif (Glu-Phe-Glu)[24]. A GFP fusion with an ER-retention signal (GFP-HDEL) localizes to the ER network and ER bodies in *A. thaliana*[8]. Onion cells that expressed GFP-HDEL revealed only the ER network (Fig. 2a, first row), while onion cells that expressed GFP-HDEL together with both BGLU23 and NAI2 revealed the ER network and a number of ER bodies (Fig. 2a, second row). Almost half of the transformed onion cells produced ER bodies (see Table 2). However, no ER bodies formed when only NAI2 or BGLU23 were expressed (Fig. 2a, third and fourth rows) or when the ER-body-membrane proteins MEB1 and MEB2 were expressed together with NAI2 (Supplementary Fig. 1)[29].

BGLU23 has an ER-retention signal (KDEL) at the C-terminus[8]. To visualize BGLU23, we used the GFP fusion protein BGLU23–GFP–KDEL, in which GFP was placed in front of the ER-retention signal. Artificial expression of BGLU23–GFP–KDEL and NAI2 in onion cells also induced the formation of fluorescent ER bodies (Fig. 2b), as did the artificial expression of BGLU23 and NAI2 (Fig. 2b). These structures had three characteristic features of *A. thaliana* ER bodies: (1) shapes similar to those of *A. thaliana* ER bodies (Fig. 2b); (2) accumulation of BGLU23 in their lumens (Fig. 2b); and (3) the ER-body-membrane protein MEB2 on

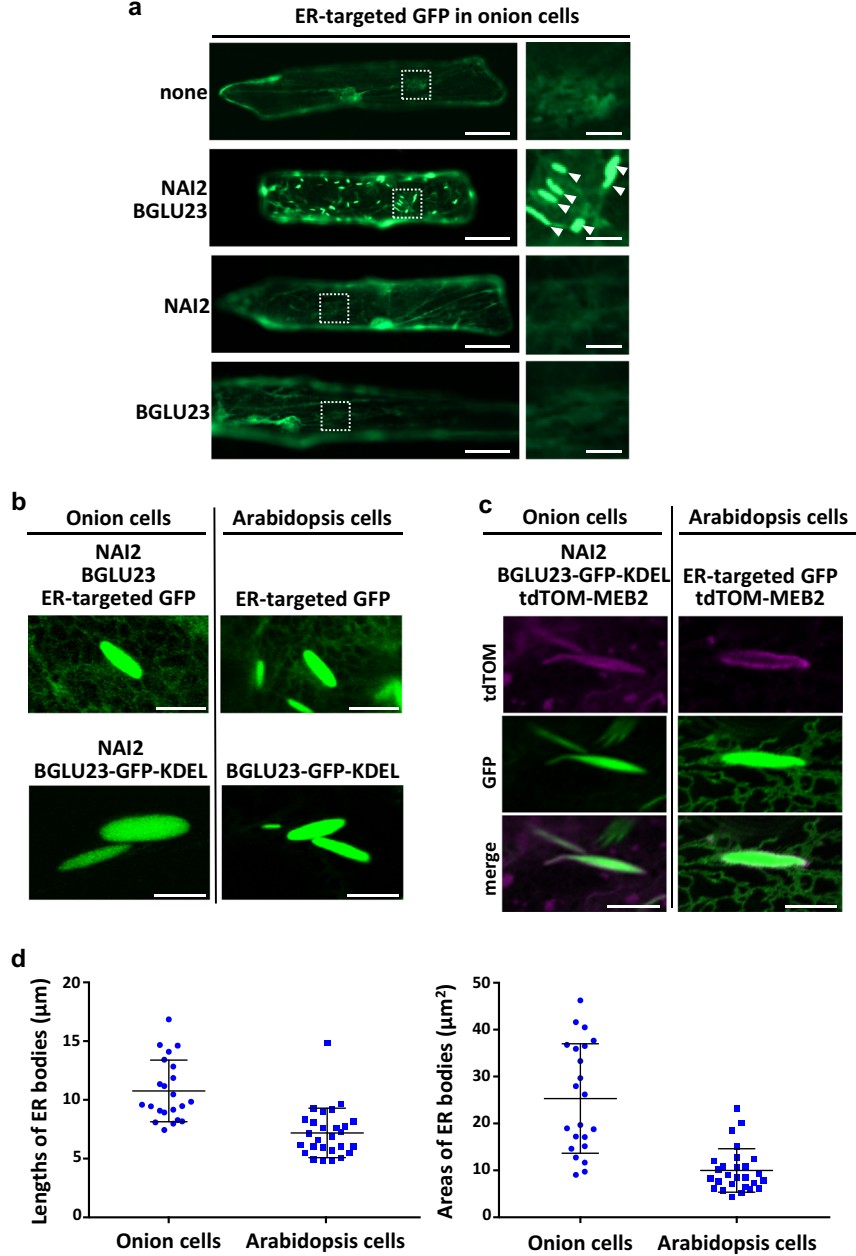

**Fig. 2 Artificial expression of the Brassicaceae-specific proteins NAI2 and BGLU23 induces the formation of ER bodies even in non-Brassicaceae plants. a** Representative fluorescence images of ER-targeted GFP in epidermal cells of onion, a non-Brassicaceae plant. ER bodies (larger than 5 μm long) are induced to form by the expression of two Brassicaceae-specific proteins (NAI2 and BGLU23), but not by the expression either protein alone. Scale bars are 50 μm for left panels and 10 μm for right panels. See Table 2 for statistical data. **b** Comparison of ER bodies in onion cells and ER bodies in *A. thaliana* cotyledon cells, both labeled with ER-targeted GFP. The onion ER bodies, like *A. thaliana* ER bodies, accumulate BGLU23–GFP–KDEL. Scale bars are 10 μm. **c** Representative fluorescence images of tdTomato-tagged ER-body-membrane protein MEB2 (tdTOM-MEB2), showing that the GFP-labeled ER bodies are surrounded with the ER-body-membrane marker MEB2. Three biological replicates were performed with similar results (see Supplementary Fig. 2). **d** Length and area of ER bodies in onion cells and *A. thaliana* cells. Four independent experiments were performed. Their lengths were 10.8 ± 2.6 μm (mean ± SD) in onion cells and 7.2 ± 2.1 μm in *A. thaliana* cells. Their areas were 25.3 ± 11.7 μm$^2$ in onion cells and 10.0 ± 4.6 μm$^2$ in *A. thaliana* cells. See Supplementary Data 3 for source data.

the surface (Fig. 2c and Supplementary Fig. 2). However, the sizes of onion ER bodies were more varied and larger than those of *A. thaliana* (Fig. 2d).

We next quantitatively evaluated the efficiency of ER-body formation in onion cells (defined as the proportion of transformed cells that produced ER bodies). Removal of the ER-retention signal of BGLU23 reduced the efficiency by roughly

two thirds (Table 2, BGLU23ΔKDEL), indicating that the ER-retention of BGLU23 promotes the formation of ER bodies. We also tried replacing BGLU23 with eight other proteins: two other β-glucosidases (myrosinases[1]), four known ER-resident proteins with ER-retention signals, and two proteins that are major components of plant ER-derived organelles (PAC vesicles[30] and KDEL vesicles[31,32]) (Table 2). Coexpression of each protein with

**Table 2 Effects of artificial expression of NAI2 in combination with another ER-synthesized protein on ER-body formation in onion cells.**

| Artificially expressed proteins | | | NAI2 | Number of cells | | Proportion of cells producing ER bodies (%) |
|---|---|---|---|---|---|---|
| Groups | Names | ER-retention signal | | Cells producing ER bodies | Transformed cells | |
| ER-body β-glucosidases | BGLU23 | Yes | − | 0 | 66 | 0 |
| | BGLU23 | Yes | + | 83 | 160 | 51.9 |
| | BGLU23ΔKDEL | No | + | 13 | 70 | 18.6 |
| Other β-glucosidases | Myrosinase (TGG4/BGLU34) | No | + | 0 | 19 | 0 |
| | Myrosinase (TGG2/BGLU37) | No | + | 1 | 103 | 1 |
| ER-resident proteins | Calreticulin (CRT1b) | Yes | + | 0 | 24 | 0 |
| | IAA-Ala conjugate hydrolase (IAR3/JR3) | Yes | + | 0 | 63 | 0 |
| | GRP94/HSP90-like (AtHSP90.7/SHD) | Yes | + | 0 | 77 | 0 |
| | Luminal binding protein (BIP2) | Yes | + | 0 | 101 | 0 |
| ER-derived organelle components | PAC-vesicle component, 12S globulin 4 (12S4) | No | + | 12 | 229 | 5.2 |
| | KDEL-vesicle component, cysteine protease (CEP3) | Yes | + | 2 | 189 | 1.1 |
| | None | − | − | 0 | 21 | 0 |
| | None | − | + | 2 | 41 | 4.8 |

Cells producing ER bodies are defined as the cells that have more than 20 ER bodies longer than 10 μm

**Table 3 Mass spectrometric data showing that BGLU23 is a candidate interacting protein with NAI2-GFP.**

| BGLU23 | GFP-h | NAI2-GFP |
|---|---|---|
| Score Sequest HT | 5 | 1660 |
| Coverage (%) | 9 | 69 |
| PSMs | 4 | 725 |
| Unique peptide numbers | 3 | 30 |
| Abundance | 3.7E+07 | 2.6E+09 |

The immunoprecipitates of NAI2-GFP-expressing seedlings (NAI2-GFP) and GFP-HDEL-expressing seedlings (GFP-h) using anti-GFP antibodies were analyzed. Score Sequest HT, sum of the scores of the individual peptides from the Sequest HT search; peptide spectrum matches (PSMs), total number of identified peptide sequences for the protein, including those redundantly identified; Abundance, sum of the associated and used peptide group abundances (= peak intensity of peptide). See Supplementary Data 2 for detailed mass spectrometric data of 80 proteins identified

NAI2 reduced the efficiency to 5.2% or less (Table 2). Thus, the combination of two Brassicaceae-specific proteins, NAI2 and BGLU23, is sufficient to trigger the formation of ER bodies in plant cells, even in non-Brassicaceae plant cells.

**Interaction between NAI2 and BGLU23.** To examine the relationship between NAI2 and BGLU23, we generated transgenic plants that expressed NAI2-GFP under control of the native *NAI2* promoter. Among the seedling proteins immunoprecipitated with anti-GFP antibodies and detected with mass spectrometry, BGLU23 had a remarkably high score (Score Sequest HT value 1660) (Table 3 and Supplementary Data 2). In addition, BGLU23 had high values on mass spectrometry, including the percent coverage of BGLU23 protein, the number of peptide spectrum matches, and the number of unique peptides) (Table 3 and Supplementary Data 2). On the other hand, the score for BGLU23 of the control transgenic plant that expressed GFP-HDEL (GFP-h)[7] was negligible (5) (Table 3 and Supplementary Data 2). These results strongly suggest that BGLU23 interacts with NAI2.

To confirm the interaction between NAI2-GFP and BGLU23, the proteins precipitated with anti-GFP antibodies were examined with an immunoblot using either anti-BGLU23 or anti-GFP antibodies (Fig. 3 and Supplementary Fig. 3). Immunoprecipitates from NAI2-GFP-expressing seedlings appeared as two major bands (Fig. 3a, IP, arrowheads), which corresponded to NAI2-GFP (Fig. 3b, IP,

anti-GFP) and BGLU23 (Fig. 3b, IP, anti-BGLU23). On the other hand, no signals were observed for immunoprecipitates from GFP-HDEL-expressing seedlings when anti-BGLU23 antibodies were used (Fig. 3b, IP, GFP-h). These results indicate that NAI2 either directly or indirectly interacts with BGLU23.

**ER bodies prevent BGLU23 from leaking into the vacuole.** ER bodies were also generated in another non-Brassicaceae plant, tobacco, by artificially expressing BGLU23–GFP–KDEL and NAI2. We established NAI2-expressing and non-NAI2-expressing tobacco cell lines, both of which stably expressed BGLU23–GFP–KDEL (Fig. 4a and Supplementary Fig. 4a). The NAI2 line formed ER bodies and trapped BGLU23–GFP–KDEL in them (Fig. 4b and Supplementary Fig. 4b, NAI2 line). On the other hand, the non-NAI2 line had no ER bodies, resulting in accumulation of BGLU23–GFP–KDEL in the vacuoles that were labeled with scopolin, a BGLU23 substrate[11] (Fig. 4b and Supplementary Fig. 4b, non-NAI2 line). These results suggest that BGLU23 is sequestered in ER bodies, to prevent it from leaking into vacuoles.

*nai2* **shows a loss of BGLU23 and a loss of resistance.** BGLU23 is the most abundant protein in *A. thaliana* roots[14]. Transgenic *A. thaliana* plants expressing BGLU23–GFP–KDEL revealed that GFP fluorescence was primarily localized to ER bodies in root hair cells (Fig. 5a and Supplementary Fig. 5a, WT), in agreement with the finding that ER bodies selectively accumulate BGLU23[8]. To examine the subcellular localization and the accumulation levels of BGLU23 in the absence of ER bodies, BGLU23–GFP–KDEL was expressed in the *A. thaliana nai2-2* mutant that has no ER bodies[33]. Despite having an ER-retention signal, BGLU23–GFP–KDEL moved from the ER into the vacuoles in *nai2-2* cells (Fig. 5a and Supplementary Fig. 5a, *nai2-2*), as was observed in the non-NAI2 tobacco cultured cell line (Fig. 4b and Supplementary Fig. 4b, non-NAI2 line). Thus, the ER-retention signal is not strong enough to retain BGLU23 in the ER.

We next examined the effect of *NAI2* deficiency on the accumulation of BGLU23 by immunoblotting analysis of root homogenates from *nai2-2* and the wild-type plants with anti-GFP and anti-BGLU23 antibodies. The levels of both

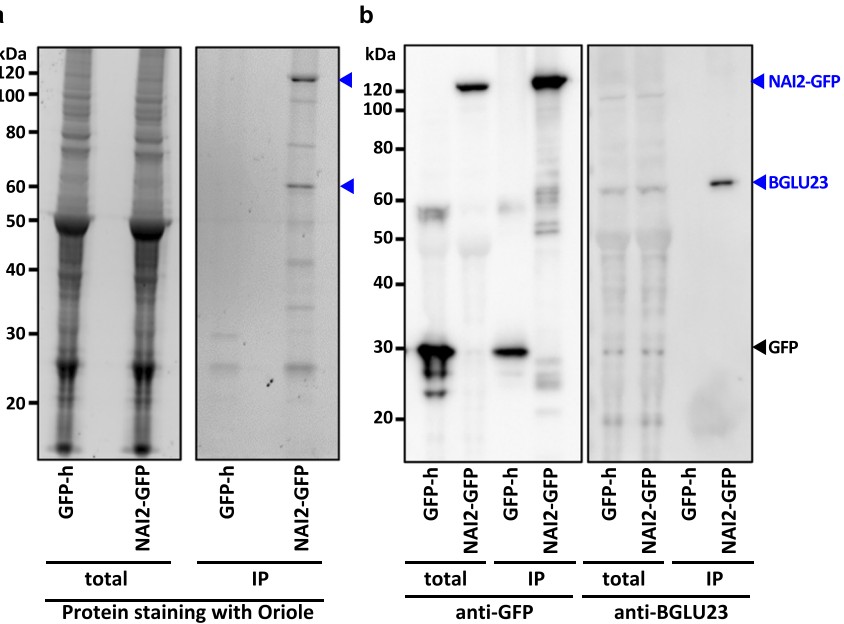

**Fig. 3 Interaction between NAI2 and BGLU23. a** Protein profiles of total homogenates (total) and immunoprecipitates with anti-GFP antibodies (IP) of 10-day-old seedlings of transgenic plants expressing GFP-HDEL (GFP-h) and transgenic plants expressing NAI2-GFP. Proteins were stained with the fluorescent dye (Oriole). Two major bands of immunoprecipitates (NAI2-GFP) are indicated by arrowheads. Two biological replicates were performed with similar results (see Supplementary Fig. 3a). **b** Immunoblots of total homogenates (total) and immunoprecipitates with anti-GFP antibodies (IP) of 10-day-old seedlings of transgenic plants expressing GFP-HDEL (GFP-h) and transgenic plants expressing NAI2-GFP using either anti-GFP antibodies or anti-BGLU23 antibodies. Two biological replicates were performed with similar results (see Supplementary Fig. 3b).

BGLU23–GFP–KDEL and endogenous BGLU23 were much lower in *nai2-2* than in the wild type (Fig. 5b and Supplementary Fig. 5b), which suggests that BGLU23 proteins that leak into the vacuoles are degraded.

To determine whether low levels of BGLU23 affect the defense against woodlice, we used a dual-choice feeding assay, in which fasted woodlice were offered wild-type and *nai2-2* roots as food. The woodlice greatly preferred the *nai2-2* roots over the wild-type roots (Fig. 5c). When fasted woodlice were given a choice between wild-type and glucosinolate-deficient mutant qKO[28] roots, they preferably ate the qKO roots (Fig. 5c). These results indicate that the BGLU23 accumulation in ER bodies is important to keep the appropriate levels for defense.

## Discussion

The present results reveal a novel chemical defense in Brassicaceae plants involving ER bodies. Our metabolomic analysis clearly shows that the endogenous substrates of ER-body β-glucosidases are various glucosinolate species. Glucosinolates are secondary metabolites that are used by the well-known myrosinase–glucosinolate herbivory defense (so called mustard-oil bomb) in Brassicaceae plants[4,5]. Both ER-body defense and myrosinase–glucosinolate defense produce repellent compounds from glucosinolates. However, ER-body defense differs from myrosinase–glucosinolate defense in that it is a single-cell type, while myrosinase–glucosinolate defense is a dual-cell (myrosin and S cells) type. Secondly, ER-body defense protects epidermal tissues most exposed to environmental stresses, while myrosinase–glucosinolate defense protects the vascular bundle of leaves[4]. Finally, ER-body defense works at the ground and underground parts of plants (seedlings and roots[34,35]), while myrosinase–glucosinolate defense works in mature leaves of the aerial parts of plants. These differences suggest that ER-body defense and myrosinase–glucosinolate defense target different types of pests and pathogens.

An essential requirement for a single-cell defense system is sequestration of β-glucosidases from their substrates. If BGLUs are not sequestered, they end up in the vacuole (Figs. 4 and 5), where they will either be degraded by lytic enzymes or react prematurely with a variety of glucoside substrates found in the vacuole. BGLU23 has a broad range of substrate specificities toward *S*-glucosides and the *O*-glucosides[11,12], and the vacuoles store a variety of glucosides[36–38]. Therefore, BGLU23 leakage into the vacuoles might cause unnecessary reaction with the glucosides.

Although the ER-retention signal KDEL does not perfectly prevent BGLU23 from leaking into the vacuoles, it contributes to the formation of ER bodies. We propose a model of a NAI2-dependent fail-safe defense system, in which BGLU23 proteins are robustly sequestered in ER bodies until they are released by tissue damage. First, the ER-retention signal primarily prevents de novo synthesized BGLU23 proteins from escaping the ER, resulting in increasing the BGLU23 levels in the ER. Second, their high levels enhance the chance of BGLU23 proteins to contact with another ER protein NAI2. Third, NAI2 helps to aggregate BGLU23 proteins in the ER subdomains, giving raise to ER bodies.

*NAI2* and homologous genes are found in Brassicaceae and Cleomaceae of the Brassicales, whose members have ER bodies in seedlings and roots[34]. The present finding that artificial expression of NAI2 and BGLU23 induces the formation of ER bodies in non-Brassicaceae species suggests that acquisition of *NAI2* by Brassicales plants was key to establishing their single-cell defense system.

## Methods

**Plant materials and growth conditions.** We used *A. thaliana* Columbia-0 (Col-0) accession. Four transgenic *A. thaliana* plants expressing GFP in ER (GFP-h), GFP-tagged BGLU23, tdTOM-MEB2 GFP-h, and NAI2-GFP were used[17,27,28,33]. Four *A. thaliana* mutants *nai1-1, nai2-2* and *bglu23-1 bglu21-1, cyp79B2 cyp79B3 myb28 myb29* (qKO, a kind gift from Barbara Ann Halkier) were used[25]. For feeding

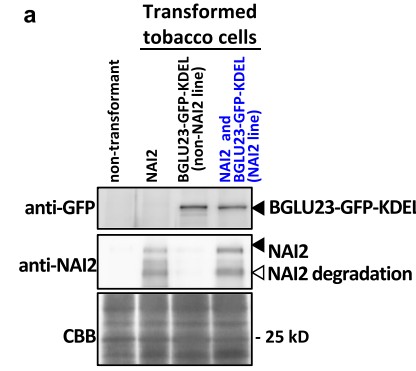

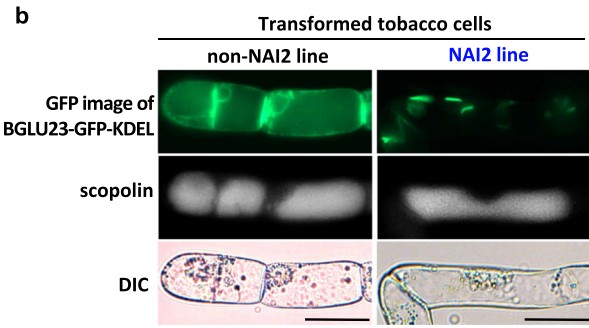

**Fig. 4 ER-body formation prevents β-glucosidase from leaking into the vacuoles. a** Immunoblots showing BGLU23–GFP–KDEL and NAI2 in four tobacco cell lines (nontransformant, transformed line with NAI2 alone, non-NAI2 line, and NAI2 line), and a Coomassie brilliant blue (CBB) staining showing a loading control. See Supplementary Fig. 4a for full images of these two immunoblots and the CBB-stained gel. **b** Representative fluorescence images of BGLU23–GFP–KDEL in NAI2 and non-NAI2 tobacco cell lines. The cells were exposed to scopoletin, leading to accumulation of scopolin, a BGLU23 substrate, in the vacuoles. Both scopoletin and scopolin are naturally fluorescent molecules. The non-NAI2 line accumulated BGLU23–GFP–KDEL in the vacuole, while the NAI2 line accumulated it in ER bodies. DIC differential interference contrast images. Scale bars are 50 μm. Three biological replicates were performed with similar results (see Supplementary Fig. 4b).

analysis, plants were germinated at 22 °C under 18 h light/6 h dark on vermiculite covered with nylon mesh sheet. For microscopic analysis, plants were germinated aseptically at 22 °C under continuous light (~100 μE•s⁻¹•m⁻²) on MS plates containing 0.4% (w/v) Gellan Gum (Wako, Tokyo, Japan), 0.5% (w/v) MES-KOH buffer (pH 5.7), and 1× Murashige and Skoog salts mixture (Wako). Onion (*Allium cepa*) bulbs were purchased from supermarket and epidermis were peeled and used for transient expression experiments as described previously[39]. Suspension cultured cells of tobacco BY-2 (*Nicotiana tabacum* cv. Bright Yellow 2) were cultured in Murashige–Skoog (MS) medium with 3% (w/v) sucrose in an orbital shaker at 140 rpm and 26 °C in dark. The cells were transferred to new medium at intervals of 1 week[40].

**Dual-choice feeding assays.** We collected woodlice (*A. vulgare*) in the Garden of Kyoto University and reared for several weeks in a breeding cage at 22 °C 18 h light/6 h dark containing a mixture of leaf soil and vermiculite. The woodlice were fed oatmeal and goldfish food for 2 or 3 weeks and starved for 2–3 days before experiments. Seeds of the wild type and one of the *A. thaliana* mutants (*bglu23 bglu21*, *nai1*, or qKO) were placed each on a 1 cm × 1 cm grid in a 9 cm diameter container with a lid and allowed to germinate for 1 week. Four replicates were done for each pair of plants. After 1 week of growth, ten adult woodlice were released in each container with a lid (no wind to avoid diffusion of volatile isothocyanates) and allowed to feed on the seedlings for 24 h. The photographs were taken before and 24 h after feeding. The cotyledon area was calculated using ImageJ (National Institute of Health) software. Alternatively, roots of aseptically grown *A. thaliana* plants (~40 mg) were placed on a wet paper in a petri dish with 9 cm diameter, and feed with three adult woodlice for 24 h. The root weight was measured before and after woodlice feeding.

**DNA construction.** To produce *GFP-tagged BGLU23* gene, we first generated a *Sal*I restriction site in BGLU23 cDNA by modifying the nucleotide sequence encoding the amino acids before the carboxyl terminal ER-retention signal. Briefly, we amplified a DNA fragment from the vector pCR8/GW/BGLU23[33] using specific primers with half of the *Sal*I site (Supplementary Table 2). The amplified DNA fragment was subsequently self-ligated to produce pCR8/GW/BGLU23-*Sal*I. The GFP gene with a *Sal*I site was inserted into the *Sal*I site of pCR8/GW/BGLU23-*Sal*I, generating the Gateway entry clone pCR8/GW/BGLU23-GFP. The protein-coding region of pCR8/GW/BGLU23-GFP was transferred to pUGW2[41] and T-vector, pK2GW7[42] to generate pUGW2/BGLU23-GFP and pK2GW7/BGLU23-GFP. These plasmids carry *Pro35S:BGLU23–GFP–KDEL*, which encodes a fusion protein of BGLU23 with GFP.

To produce *tdTom-MEB2* fusion gene, the entire protein-coding region of pENTR/MEB2 plasmid[29] was transferred to pB4tdGW vector (a kind gift from Shoji Mano, National Institute for Basic Biology) to generate pB4td/MEB2. The plasmid carries *Pro35S:tdTom-MEB2*, which encodes fusion proteins of tdTomato with MEB2 in T-DNA region. The detail of ptd/MEB2 plasmid harboring *Pro35S: tdTom-MEB2* for particle bombardment is descried in Yamada et al.[29].

*A. thaliana* cDNAs encoding At3g15950 (NAI2) was amplified by RT-PCR using gene-specific primers (Supplementary Table 2) and cloned into the Gateway entry vector pENTR/SD/D (Invitrogen) to generate pENTR/SD/D/NAI2. The protein-coding region with 5′UTR of pENTR/SD/D/NAI2 was transferred to pUGW2[41] and T-vector, pH2GW7[42] to generate pUGW2/NAI2 and pH2GW7/NAI2. These plasmids carry *Pro35S:NAI2*.

To produce NAI2-GFP, the genomic sequence of *A. thaliana* NAI2 and the backbone sequence of pENTR1A vector (Invitrogen) were amplified using specific primers (Supplementary Table 2), then these two fragments were combined by the In-Fusion reaction (Takara Clontech). The promoter sequence of *NAI2* was amplified using specific primers with Gateway attB adaptor sequences and cloned into pDONR P4-P1r vector (Invitrogen) by the Gateway BP reaction. The resulting entry vectors, pENTR1A/NAI2gΔstop and pDONR P4-P1r/ProNAI2, were reacted with R4pGWB504[43] by the Gateway LR reaction (Invitrogen) to generate R4pGWB504/ProNAI2:NAI2g-sGFP vector.

*A. thaliana* cDNAs encoding At5g25980 (BGLU37) was amplified using gene-specific primers (Supplementary Table 2) and cloned into the Gateway entry vectors pENTR/D (Invitrogen). *A. thaliana* cDNAs encoding At1g09210 (CRT1b), At1g51760 (IAR3), At3g48350 (CEP3), At4g24190 (HSP90.7), and At5g42020 (BIP2) were amplified using gene-specific primers (Supplementary Table 2), and cloned into the Gateway entry vector pCR8/GW (Invitrogen). A cDNA clone in the Gateway entry vector, U25668 (At1g47600, BGLU34), was obtained from the Arabidopsis Biological Resource Center. The cDNA clone of At1g47600 (12S4) in the Gateway entry vector was kindly provided by Masatake Kanai (National Institute for Basic Biology). The entire protein-coding region in the Gateway entry vector was transferred into pUGW2 (31), which contains the 35S promoter to generate pUGW2/BGLU37, pUGW2/CRT1b, pUGW2/IAR3, pUGW2/CEP3, pUGW2/HSP90.7, pUGW2/BIP2 pUGW2/BGLU34, and pUGW2/12S4. The pBI221/SP-GFP-HDEL plasmid harboring *Pro35S:SP-GFP-HDEL* was kindly provided by Kentaro Tamura (Kyoto University).

**Generation of transgenic plants and culture cells.** The plasmid pK2GW7/BGLU23-GFP was introduced into *Agrobacterium tumefaciens* strain C58C1Rif and then transformed into *A. thaliana* plants. The same *Agrobacterium* was used to transform BY-2 culture cells. Transformed cells that are resistance to kanamycin were selected. The plasmid pB4td/MEB2 was introduced into *Agrobacterium* strain GV3101 (pMP90RK) and then transformed into *A. thaliana* GFP-h plants to generate td-MEB2 GFP-h. The plasmid R4pGWB504/ProNAI2:NAI2g-sGFP was introduced into *Agrobacterium* strain GV3101 and then transformed into *A. thaliana nai2-2* plants to generate *ProNAI2:NAI2g-sGFP/nai2-2*. The plasmid pH2GW7/NAI2 was introduced into *Agrobacterium tumefaciens* strain GV3101 and then transformed into BY-2 culture cells. Transformed cells that are resistance to hygromycin were selected.

**Scopolin production in tobacco cells.** Five-day-old BY-2 cell suspension (3 ml) was treated with 15 μl of 0.1 M scopoletin (Sigma) in dimethyl sulfoxide to produce and accumulate scopolin in vacuole[44].

**Particle bombardment.** The plasmids for particle bombardment (pUGW2/XXX, ptd/MEB2, and pBI221/SP-GFP-HDEL) were bombarded into onion epidermis cells using the Biolistic Particle Delivery System (Bio-Rad Laboratories) according to the manufacturer's instructions.

**Immunoprecipitation.** Ten-day-old seedlings of GFP-h and *ProNAI2:NAI2g-sGFP/nai2-2* were ground with three-time volume of extraction buffer [50 mM Tris-HCl (pH 8.0), 150 mM NaCl, 1% Triton X-100, protein inhibitor (complete, EDTA-free; Roche), and 1 mM phenylmethylsulfonyl fluoride (PMSF)] to fresh weight. The extracts were centrifuged at 20,000 g for 10 min at 4 °C. The supernatants (1.5 mL each) were subjected to immunoprecipitation with 50 μL of anti-GFP microbeads (μMACS GFP tag protein isolation kit, Milteny Biotech), and then

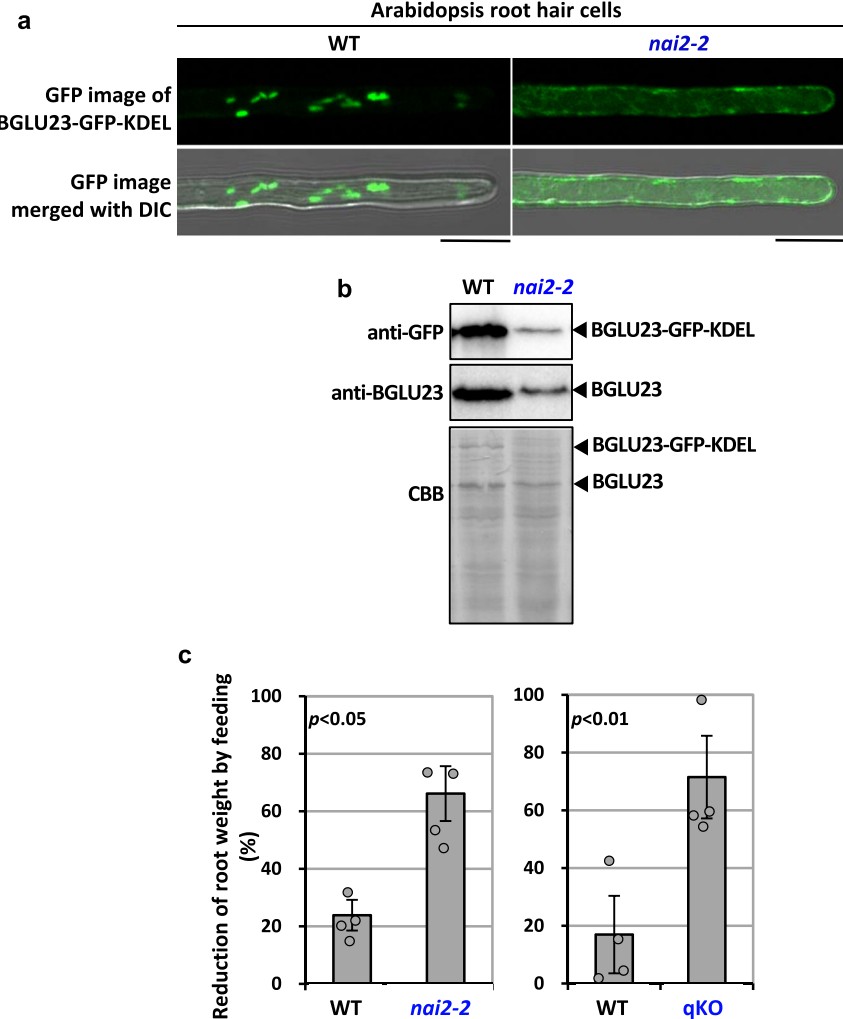

**Fig. 5 NAI2 is required for accumulation of BGLU23 in ER bodies and for chemical defense against predators. a** Representative fluorescence images of BGLU23–GFP–KDEL in *A. thaliana* root hair cells, showing that BGLU23–GFP–KDEL is accumulated in ER bodies of the wild type (WT), while it is accumulated in the vacuole of *nai2-2*. DIC, differential interference contrast images. Scale bars are 20 μm. Two biological replicates were performed with similar results (see Supplementary Fig. 5a). **b** Accumulation of BGLU23–GFP–KDEL and endogenous BGLU23 in 3-week-old wild type (WT) and *nai2-2*. Immunoblot with anti-GFP antibody shows BGLU23–GFP–KDEL protein level. Immunoblot with anti-BGLU antibody shows endogenous BGLU23 protein level. Coomassie brilliant blue (CBB) staining as a loading control. See Supplementary Fig. 5b for full images of these two immunoblots and the CBB-stained gel. **c** Effects of NAI2 deficiency on woodlouse feeding on roots of *A. thaliana* mutants: *nai2-2* and the glucosinolates-deficient quadruple mutant qKO. Reduction of root weight by feeding is shown. Error bars indicate standard error of three independent experiments. Significance values were calculated by two-sided Student's *t* test. See Supplementary Data 3 for source data.

were applied to a column (μ column, Milteny Biotech). Pure immunoprecipitates were eluted with 70 μL of 2× SDS sample buffer.

**Immunoblot**. Immunoblot was performed as described previously[33]. Total protein was extracted from 100 μl 7-day-old BY-2 cells or 50 mg of *A. thaliana* roots with 200 μl 2× sample buffer [20 mM Tris-HCl buffer, pH 6.8, 40% (v/v) glycerol, 2% (w/v) SDS, and 2% (v/v) 2-mercaptethanol]. The extract (10 μl) was subjected to SDS-PAGE, transferred to a nylon membrane, and then subjected to immunoblot analysis using anti-GFP (1:2000 dilution), anti-BGLU23/IM (1:5000 dilution)[45], and anti-NAI2/ΔSP (1:2000 dilution)[33]. Alternatively, the proteins were stained with Coomassie Brilliant Blue as a loading control. For immunoprecipitation analysis, the supernatant and elusion were used as total protein and bound fractions, respectively. Twenty microliters of each fraction was subjected to SDS-PAGE, transferred to a PVDF membrane, and then subjected to immunoblot analysis using anti-GFP (1:1000 dilution) and anti-BGLU23/IM (1:2000 dilution). Alternatively, the proteins were stained with Oriole Fluorescent Gel Stain (Bio-Rad).

**Protein mass spectroscopy**. Samples were analyzed by nano-flow reverse-phase liquid chromatography followed by tandem MS, using a Q Exactive Hybrid Quadrupole-Orbitrap Mass Spectrometer (Thermo Fisher Scientific) as described[46]. A capillary reverse-phase HPLC-MS/MS system was composed of a Dionex

U3000 gradient pump equipped with a VICI CHEMINERT valve, and a Q Exactive equipped with a Dream Spray nano-electrospray ionization (NSI) source (AMR, Tokyo, Japan). Samples were automatically injected using a PAL System auto-sampler (CTC Analytics, Zwingen, Switzerland) and a peptide L-trap column (Trap and Elute mode, Chemical Evaluation Research Institute, Tokyo) attached to an injector valve for desalinating and concentrating peptides. After washing the trap with MS-grade water containing 0.1% (v/v) trifluoroacetic acid and 2% (v/v) acetonitrile (solvent C), the peptides were loaded onto a separation capillary C18 reverse-phase column (NTCC-360/100–3–125, 125 × 0.1 mm, Nikkyo Technos, Tokyo). The eluents used were: A, water containing 0.5% (v/v) acetic acid, and B, 80% (v/v) acetonitrile containing 0.5% (v/v) acetic acid. The column was developed at a flow rate of 0.5 μL min⁻¹ with an acetonitrile concentration gradient of 5% B to 40% B for 100 min, then 40% B to 95% B for 1 min, 95% B for 3 min, 95% B to 5% B for 1 min, and finally reequilibrating with 5% B for 10 min. Xcalibur 3.0.63 (Thermo Fisher Scientific) was used to record peptide spectra over a mass range of m/z 350–1800. MS spectra were recorded followed by ten data-dependent high-energy collisional dissociation (HCD) MS/MS spectra generated from the ten highest intensity precursor ions. Multiply-charged peptides were chosen for MS/MS experiments due to their good fragmentation characteristics. MS/MS spectra were interpreted and peak lists were generated using Proteome Discoverer 2.0.0.802 (Thermo Fisher Scientific). Searches were performed using SEQUEST (Thermo Fisher Scientific) against the *A. thaliana* (TAIR TaxID = 3702) peptide sequence

database. Search parameters were set as follows: enzyme selected with two maximum missing cleavage sites, a mass tolerance of 10 ppm for peptide tolerance, 0.02 Da for MS/MS tolerance, fixed modification of carbamidomethyl (C), and variable modification of oxidation (M). Peptide identifications were based on a significant Xcorr values (high confidence filter). Peptide identification and modification information returned from SEQUEST were manually inspected and filtered to obtain confirmed peptide identification and modification lists of HCD MS/MS.

**Microscopy.** A confocal laser scanning microscope (LSM510, Carl Zeiss) was used to observe fluorescent proteins. An argon laser (488 nm) and a 505/530 nm band-pass filter were used to observe GFP, and a helium–neon laser (543 nm) and a 560/615 nm band-pass filter were used to observe tdTomato. An epifluorescence microscope (Axioimager Z1, Carl Zeiss) was used to observe fluorescent proteins and scopolin in onion and tobacco cells.

**SDS-PAGE and in-gel digestion of protein samples.** SDS-PAGE was carried out according to the method described by Laemmli[47]. Protein samples were partially separated on a slab gel (~1 cm). In-gel digestion was performed essentially as described by Rosenfeld[48]. Each lane was excised and was chopped into small pieces. Proteins in the gel pieces were digested for 16 h at 37 °C in a reaction buffer (50 mM ammonium bicarbonate buffer, pH 8.0) containing 0.01 mg/mL trypsin (Promega).

**Metabololmic analysis.** Seven-day-old *A. thaliana* seedlings were homogenized in liquid nitrogen and subjected to LC–MS/MS analysis by the outsourcing from Kazusa DNA Research Institute (LC–MS Basic Analysis, http://www.biosupport.kazusa.or.jp/sub_center3/index.php/lcms-basic/). The obtained data were statistically analyzed with R software to find out the differentially accumulated chemicals in each sample. The data were deposited to MetaboLight[49].

**Statistics and reproducibility.** In both dual-choice feeding assay and metabolomic analysis, three to four independent experiments were performed to obtain mean ± SE. In ER-body size analysis, four independent experiments were performed to obtain mean ± SD. The statistical analysis was conducted with two-sided Student's *t* test.

**Reporting summary.** Further information on research design is available in the Nature Research Reporting Summary linked to this article.

## Data availability

Nucleotide and protein sequence data are available from the GenBank/EMBL and UniProt database with following IDs: *NAI1*/At2g22770 (Gene ID, 816807; UniProt entry, Q8S3F1), *NAI2*/At3g15950 (820839; Q9LSB4), *PYK10/BGLU23*/At3g09260 (820082; Q9SR37), *BGLU21*/At1g66270 (842944; Q9C525), *MEB1*/At4g27860 (828899; Q8W4P8), and *MEB2*/At5g24290 (832496; F4KFS7). Proteome data were deposited in PRIDE with accession number (PXD016606). Metabolome data were deposited in MetaboLights with accession number (MTBLS1383). All other data supporting the findings of this study and newly generated plasmids in this study are available from the corresponding author upon reasonable request.

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

## Acknowledgements

We are grateful to Hideharu Numata (Kyoto University) for his instruction of breeding woodlice; to Barbara Ann Halkier (University of Copenhagen) and the Arabidopsis Biological Resource Center for providing *A. thaliana* mutant seeds; to Masatake Kanai (NIBB), Shoji Mano (NIBB), and Kentaro Tamura (Kyoto University) for their donation of the vectors; to Momoko Nishina (NIBB) and Yumi Yoshinori (NIBB) for their technical assistance; to the Model Plant Research Facility (NIBB BioResource Center) for the technical supports: and to James Raymond (Eigoken) for critical readings of this paper. This work was supported; by OPUS12 to K.Y. (no. UMO-2016/23/B/NZ1/01847) from National Science Center Poland; by Grants-in-Aid for Scientific Research to I.H.-N. (no. 15H05776) and to M.N. (no. 1685209) from the Japan Society for the Promotion of Science (JSPS); by Specially Promoted Research of Grant-in-Aid for Scientific Research to I.H.-N. (no. 22000014) from JSPS; and by the Hirao Taro Foundation of KONAN GAKUEN for Academic Research to I.H.-N.

## Author contributions

K.Y., M.N. and I.H-.N conceived the study. K.Y. designed the experiments: S.G.-Y. and K.Y. performed particle bombardment and microscopic observation and A.N. and K.Y. performed dual-choice feeding assays. A.J.N. conducted statistical analysis of metabolome data. T.K. performed pull-down experiments and K.K. performed proteome experiments. K.Y. and I.H-.N analyzed whole data and wrote the paper.

## Competing interests

The authors declare no competing interests.
