## [Peer Review File · Communications Biology]

Reviewers' comments:

Reviewer #1 (Remarks to the Author):

Endoplasmic reticulum-derived bodies enable a single-cell chemical defense in Brassicaceae plants: by Yamada et al.

This manuscript reports an important single-cell defense mechanism in Arabidopsis, and should be published in Communications Biology.

I have several minor comments.

1. Please show the number of replications of the experiments throughout the manuscript.
2. Abstract and seventh paragraph. To discuss of the possibility of the single-cell chemical defense against microorganisms, the authors should conduct the experiments using plant pathogens.
3. The first paragraph and Figure 1abc. Figure 1abc show an example. It would be convincing to show the data and statistics: e.g. length and area of ER-body like structure and ER-body. What do you mean by ER body-like structures in Figure 1a?
4. The second paragraph and Figure 1d. Figure 1d is not a figure. Please be specific to explain Figure 1d in the text: what do you mean by "diminish", "strongly" etc.
5. The authors use onion and tobacco. Why not one plant species?
6. The third paragraph and Figure 1e. Again, Figure 1e is an example.
7. The fourth paragraph and Figure 2bc. Again, Figure 2b is an example. Figure 2c: conversion rates. Here mean and SE, and statistics are needed.
8. The fifth paragraph and Table 2. Though apparent, the authors should conduct the statistical analyses (no need to compare with ns). Some of the data may show the significant difference depending on the number of the replication.
9. The sixth paragraph, method and Figure 3. Detailed explanation of the bioassay is needed: Where the woodlice (or and woodlouse) were released? How many replicates? Wind direction if any. It would be nice to show the real data rather than relative damage (%) and analyze them with appropriate statistics. the value of -20 in Figure 3 is bit strange.
Wood louses should be woodlice (method).
Better to say adult woodlice
As Armadillidium vulgare is shown in the abstract section, no need to repeat in the text.
" because they normally feed on detritus, i.e., avoid fresh seedlings." If so, the reference is needed, otherwise delete.
The last sentence. against feeding damage  against adult woodlice
10. The last paragraph. Lines 4 from the last to the end. Too hypothetical.

Reviewer #2 (Remarks to the Author):

The manuscript written by Yamada reported that endoplasmic reticulum-derived bodies play an important role in a single-cell chemical defense in Brassicaceae plants. The authors found that the combination of two Brassicaceae-specific proteins, NAI2 and BGLU23, is sufficient to trigger the formation of ER bodies in plant cells, including non-Brassicaceae plant cells. Moreover, they found that ER bodies sequester BGLU23 and this is important for ER body formation. Finally, they observed that BGLU23 and BGLU21 function as glucosinolate-converting β -glucosidases and that the herbivore preferred to feed on Arabidopsis mutants (bglu23bglu21, nai1 or qKO) over on WT.

Main concerns:

1)P3, the last sentence in the first paragraph: how can the authors say "BGLU23 is collected into ER bodies efficiently in an NAI2-dependent manner"? The result only showed that BGLU23 targeted into ER is important for ER body formation.

2)P3, the last sentence in the second paragraph: how can the authors say "these results indicate that the Brassicaceae-specific protein NAI2 has a pivotal role in sequestering BGLU23 in ER bodies in plant cells"? The result only showed that ER bodies sequester BGLU23.

3)P4, the second paragraph: how can the authors concluded "These results indicate that BGLU23 and BGLU21 function as major glucosinolate-converting β -glucosidases of seedlings". You do not know how many glucosinolates were totally converted.

4)Are there ER bodies in *bglu23bglu21* mutants? When plants infested by the herbivore, BGLU23 and 21 can move to vacuoles and then produce toxic compounds?

5)What are toxic compounds from glucosinolates in Arabidopsis? Are there lower levels in toxic compounds in mutants (*bglu23bglu21*, *nai1* or *qKO*) after herbivore infestation?

6)Are there higher levels in toxic compounds in *nai2-2* mutant? And it showed higher resistance to the herbivore than did WT plants?

Reviewer #3 (Remarks to the Author):

The authors have investigated the formation of ER bodies in plants. Artificial co-expression of BGLU23 and NAI2 induced ER body-like structures in onion, a non-Brassicaceae plant. Removal of the ER retention signal at the C-termini of BGLU23 failed to form ER bodies. In addition, expression of other ER-located proteins with NAI2 failed to strongly induce ER-bodies in onion cells. The BGLU23-GFP-KDEL located into the vacuoles in the ER body-deficient mutant, *nai2*. Further experiments using NAI2-expressing or non-NAI2-expressing lines confirmed the role of NAI2 in sequestering BGLU23 to the ER-bodies. The authors profiled the metabolites in WT and *bglu23/bglu21* mutant before and after incubation at 26 °C. 76 metabolites were decreased during incubation in a beta-glucosidase-dependent manner. Of which, 13 were glucosinolates and they were disappeared after incubation in WT but not disappeared in *bglu23/bglu21* homogenate. The authors also demonstrated that woodlice selectively ate the mutants with defects in the synthesis of glucosinolates. The authors concluded that NAI2 is a regulator that is essential for the generation of ER bodies and the ER bodies are responsible for a robust single-cell type of chemical defense against predation. Quite how NAI2 together with BGLU23 form the ER body is not known (or determined), but the findings are interesting and its reasonable that this is left for future study.

Major points:

The data supporting NAI2 and BGLU23 as essential components for generating ER-bodies. However, no attempt to determine if co-expression of BGLU21 or BGLU23 and NAI2 or NAI1 forms ER bodies. This point should be included.

The results also showed that BGLU23 is located in the ER bodies and relocated into the vacuoles in *nai2* mutant. The authors suggest that NAI2 sequestered the BGLU23 to the ER bodies but this is not at all clear and the results supporting the conclusion is weak, given that *nai2* is a ER body-deficient

mutant. Whether NAI2 or ER body deficiency leads to the mistargeting of BGLU23 needs to be clarified and sufficiently discussed.

The results support that BGLU23 and BGLU21 are major glucosidases generating glucosinolates through metabolome profiling with WT and *bglu23/bglu21* mutant. The effects of *bglu23* and *bglu21* single mutant on metabolome and herbivore damage need to be included in the analysis. Also please explain why not include BGLU21 in the previous experiments.

Minor points:

Figure 2c, please add statistical analysis here and indicate how many biological replicates.

Figure 3, indicate whether they are one- or two-sided Student t-test.

Table 1 and Table S1, please add statistical analysis.

Responses to Reviewers

Reviewer #1 (Remarks to the Author):

Endoplasmic reticulum-derived bodies enable a single-cell chemical defense in Brassicaceae plants: by Yamada et al.

This manuscript reports an important single-cell defense mechanism in Arabidopsis, and should be published in Communications Biology.

I have several minor comments.

1. Please show the number of replications of the experiments throughout the manuscript.

Done.

2. Abstract and seventh paragraph. To discuss of the possibility of the single-cell chemical defense against microorganisms, the authors should conduct the experiments using plant pathogens.

We showed that the defense system was effective against woodlice and implied that it was also effective against microorganisms, but didn't show it. In response to this comment, we infected seedlings with several pathogenic fungi and one pathogenic bacterium (*Pseudomonas syringae*). For the fungi, we infected 7-day-old seedlings and roots of *bglu23 bglu21, nai1*, and the wild type on agar plates with spore suspensions of fungi (*Colletotricum higginsianum, Alternaria brassiciola, Blumeria graminis, Fusarium oxysporum* and *Pythium sylvaticum*). The infection procedures were done essentially as described (Shimada et al., 2014; Hatsugai et al., 2009).

We observed no significant differences in lesion formations or plant growth between the mutants and the wild type after infection of 1-5 days. So as a result, we deleted the statements about microorganismal defense in the Abstract and Discussion.

3a. The first paragraph and Figure 1abc. Figure 1abc show an example. It would be convincing to show the data and statistics: e.g. length and area of ER-body like structure and ER-body.

Note that Fig. 1 is now Fig. 2. We changed the legend from "Fluorescence images" to "Representative fluorescence images" here and in other figures as well. For each of the four panels in Fig. 2a, we provided quantitative data (the proportion of transformed cells that produced ER bodies) in a new table (Table 2). For Fig. 2b, we added new panels (Fig. 2d) that give statistical data on the lengths and areas of ER bodies in onion and Arabidopsis cells. For each of the images in Fig. 2c, we provided two additional examples in Supplementary Fig. 2.

3b. What do you mean by ER body-like structures in Figure 1a?

To clarify the structures in the panel, we added magnified views and indicated a number of ER bodies by arrowheads (Fig. 2a, second row, right panel). These structures had three characteristic features of Arabidopsis ER bodies: 1) shapes similar to those of Arabidopsis ER bodies (Fig. 2b), 2) accumulation of BGLU23 in their lumens (Fig. 2b, lower), and 3) the ER-body-membrane protein MEB2 on the surface (Fig. 2c and supplementary Fig. 2), although the sizes of onion ER bodies are more varied and larger than those of Arabidopsis ER bodies (Fig. 2d). To avoid confusion, we changed “ER body-like structures” to “ER bodies” throughout the manuscript.

4. The second paragraph and Figure 1d. Figure 1d is not a figure. Please be specific to explain Figure 1d in the text: what do you mean by "diminish", "strongly" etc.

We replaced Fig. 1d with a new table (Table 2).

We changed [Loss of ER retention signal] “diminished the ability of BGLU23 to form ER bodies” to “Removal of the ER-retention signal of BGLU23 reduced the proportion of cells producing ER bodies by roughly two thirds” (Table 2).

We changed “failed to strongly induce ER-body formation” to “Coexpression of each protein with NAI2 reduced the proportion of cells producing ER bodies to 5.2% or less” (Table 2).”

5. The authors use onion and tobacco. Why not one plant species?

We want to show the ability of the ectopic expression of a combination of NAI2 and BGLU23 to induce formation of ER bodies in non-Brassicacea plants including both monocot and dicot plants. We selected onion (a monocot) and tobacco (a dicot) because their transformation technologies are well established.

6. The third paragraph and Figure 1e. Again, Figure 1e is an example.

Note that Fig. 1e is now Fig. 5a. We changed the legend to “Representative fluorescence images...” and provided additional photos in Supplementary Fig. 5.

7a. The fourth paragraph and Figure 2bc. Again, Figure 2b is an example.

Note that Fig. 2b is now Fig. 4b. We changed the legend to “Representative fluorescence images...” and provided additional photos in Supplementary Fig. 4.

7b. Figure 2c: conversion rates. Here mean and SE, and statistics are needed.

To do replicate experiments, we needed multiple transformed cell lines that have stable expression levels of NAI2. But the NAI2 expression levels varied among the lines. Because of the instability, we decided that the rate data was not reliable enough to report and deleted it.

8. The fifth paragraph and Table 2. Though apparent, the authors should conduct the statistical analyses (no need to compare with ns). Some of the data may show the significant difference depending on the number of the replication.

We think the reviewer is referring to Table 1 here. Before and after signals that are significantly different ($p < 0.05$; two-sided Student's *t*-test) are marked with asterisks.

9a. The sixth paragraph, method and Figure 3. Detailed explanation of the bioassay is needed: Where the woodlice (or and woodlouse) were released? How many replicates? Wind direction if any.

the value of -20 in Figure 3 is bit strange.

Wood louses should be woodlice (method).

Better to say adult woodlice

As *Armadillidium vulgare* is shown in the abstract section, no need to repeat in the text.

"because they normally feed on detritus, i.e., avoid fresh seedlings." If so, the reference is needed, otherwise delete.

The last sentence. against feeding damage  against adult woodlice

Note that Fig. 3 is now Fig. 1. We revised the methods section as follows.

Dual-choice feeding assays

Woodlice (*Armadillidium vulgare*) were collected in the Garden of Kyoto University and reared for several weeks in a breeding cage at 22 °C 18 h light / 6 h dark containing a mixture of leaf soil and vermiculite. The woodlice were fed oatmeal and goldfish food for 2 or 3 weeks and starved for 2 to 3 days before experiments. Seeds of the wild type and one of the *A. thaliana* mutants (*bglu23 bglu21*, *nai1*, or qKO) were placed each on a 1 cm x 1 cm grid in a 9-cm diameter container with a lid and allowed to germinate for 1 week. Four replicates were done for each pair of plants. After 1 week of growth, 10 adult woodlice were released in each container with a lid (no wind to avoid diffusion of volatile isothiocyanates) and allowed to feed on the seedlings for 24 h. The photographs were taken before and 24 h after feeding. The cotyledon area was calculated using ImageJ (National Institute of Health) software. Alternatively, roots of aseptically grown *A. thaliana* plants (~40 mg) were placed on a wet paper in a petri dish with 9-cm diameter, and feed with three adult woodlice for 24 h. The root weight was measured before and after woodlice feeding.

9b. It would be nice to show the real data rather than relative damage (%) and analyze them with appropriate statistics.

It is difficult to show real feeding damage (absolute change in cotyledon area) because the cotyledons of the mutants were almost completely eaten after 24 hr. This means that the area damaged by feeding corresponds to the starting cotyledon area, which varied among the experiments.

10. The last paragraph. Lines 4 from the last to the end. Too hypothetical.

We changed the original sentence:

“The finding that artificial expression of NAI2 induces the formation of ER bodies in non-Brassicaceae species raises the possibility that acquisition of the NAI2 gene by a common ancestor of the Brassicaceae and Cleomaceae led to their novel single-cell defense system.”

to

“The present finding that artificial expression of NAI2 and BGLU23 induces the formation of ER bodies in non-Brassicaceae species suggests that acquisition of *NAI2* by Brassicales plants was key to establishing their single-cell defense system.”

Reviewer #2 (Remarks to the Author):

The manuscript written by Yamada reported that endoplasmic reticulum-derived bodies play an important role in a single-cell chemical defense in Brassicaceae plants. The authors found that the combination of two Brassicaceae-specific proteins, NAI2 and BGLU23, is sufficient to trigger the formation of ER bodies in plant cells, including non-Brassicaceae plant cells. Moreover, they found that ER bodies sequester BGLU23 and this is important for ER body formation. Finally, they observed that BGLU23 and BGLU21 function as glucosinolate-converting β -glucosidases and that the herbivore preferred to feed on *Arabidopsis* mutants (bglu23bglu21, nai1 or qKO) over on WT.

Main concerns:

1)P3, the last sentence in the first paragraph: how can the authors say “BGLU23 is collected into ER bodies efficiently in an NAI2-dependent manner”? The result only showed that BGLU23 targeted into ER is important for ER body formation.

We changed the sentence to

“Thus, the combination of two Brassicaceae-specific proteins, NAI2 and BGLU23, is sufficient to trigger the formation of ER bodies in plant cells....”

2)P3, the last sentence in the second paragraph: how can the authors say “these results indicate that the Brassicaceae-specific protein NAI2 has a pivotal role in sequestering BGLU23 in ER bodies in plant cells”? The result only showed that ER bodies sequester BGLU23.

This sentence was based on our findings that BGLU23 moved to the ER bodies in the presence of NAI2 and moved to the vacuoles in the absence of NAI2. In the revised manuscript, we removed the sentence.

Instead, we added the statement “NAI2 helps to aggregate BGLU23 proteins in the ER subdomains, giving raise to ER bodies.” in Discussion.

The statement is based on the additional data showing the interaction between NAI2 and BGLU23 (Fig. 3, Supplementary Fig. 3, and Supplementary Data 1) the NAI2 requirement for accumulation of BGLU23 and for defense against Woodlice (Fig. 5a and 5c; Supplementary Fig. 5b).

3)P4, the second paragraph: how can the authors concluded “These results indicate that BGLU23 and BGLU21 function as major glucosinolate-converting β -glucosidases of seedlings”. You do not know how many glucosinolates were totally converted.

Glucosinolate species in *A. thaliana* seedlings were reported (Variation of glucosinolate accumulation among different organs and developmental stages of *Arabidopsis thaliana*).

Phytochemistry, 62: 471-81, 2003). All of these glucosinolates were detected in our comparative metabolomic analysis (Supplementary Table 1) and each of their levels was reduced in a BGLU23/BGLU21-dependent manner (Table 2). We cite this paper in the revised manuscript.

4a) Are there ER bodies in *bglu23bglu21* mutants?

Yes, we previously showed that *bglu23bglu21* mutants have ER bodies, although the ER bodies have irregular shapes (Nagano et al. 2009).

4b) When plants infested by the herbivore, BGLU23 and 21 can move to vacuoles and then produce toxic compounds?

Herbivore infesting damages the organelles, which brings BGLU23 and BGLU21 into contact with substrates that were stored in the vacuoles and then produces toxic compounds.

5a) What are toxic compounds from glucosinolates in Arabidopsis?

They are isothiocyanates (ITCs). We added it in Introduction by citing references.

5b) Are there lower levels in toxic compounds in mutants (*bglu23bglu21*, *nai1* or *qKO*) after herbivore infestation?

Isothiocyanates (ITCs) are highly reactive and easily conjugate with other chemicals, which makes it difficult to measure their levels. Most of ITCs are volatile. We tried to directly measure their levels by Gas chromatography-mass spectrometry, but have failed it.

Instead, we can estimate the ITC levels, based on the concept that high levels of β -glucosidases (enzymes) and/or glucosinolates (substrates) reflect high levels of ITCs (products). In this concept, the three mutants have much lower ITC levels because (1) *bglu23bglu21* has no glucosinolate-converting activities (Table 1), (2) *nai1* has neither BGLU32 nor BGLU21, and (3) *qKO* has no glucosinolates.

6) Are there higher levels in toxic compounds in *nai2-2* mutant? And it showed higher resistance to the herbivore than did WT plants?

We compared *nai2-2* and the wild type with respect to 1) BGLU23 in root cells, 2) BGLU23 levels, and 3) woodlice feeding. The results showed that deficiency of NAI2 causes BGLU23 to leak into the vacuole where it is degraded and a loss of resistance to woodlice. Thus, *nai2-2* has lower ITC levels and lower resistance to woodlice. We added the results in a new Fig. 5.

Reviewer #3 (Remarks to the Author):

The authors have investigated the formation of ER bodies in plants. Artificial co-expression of BGLU23 and NAI2 induced ER body-like structures in onion, a non-Brassicaceae plant. Removal of the ER retention signal at the C-termini of BGLU23 failed to form ER bodies. In addition, expression of other ER-located proteins with NAI2 failed to strongly induce ER-bodies in onion cells. The BGLU23-GFP-KDEL located into the vacuoles in the ER body-deficient mutant, *nai2*. Further experiments using NAI2-expressing or non-NAI2-expressing lines confirmed the role of NAI2 in sequestering BGLU23 to the ER-bodies. The authors profiled the metabolites in WT and *bglu23/bglu21* mutant before and after incubation at 26 °C. 76 metabolites were decreased during incubation in a beta-glucosidase-dependent manner. Of which, 13 were glucosinolates and they were disappeared after incubation in WT but not disappeared in *bglu23/bglu21* homogenate. The authors also demonstrated that woodlice selectively ate the mutants with defects in the synthesis of glucosinolates. The authors concluded that NAI2 is a regulator that is essential for the generation of ER bodies and the ER bodies are responsible for a robust single-cell type of chemical defense against predation. Quite how NAI2 together with BGLU23 form the ER body is not known (or determined), but the findings are interesting and its reasonable that this is left for future study.

Major points:

The data supporting NAI2 and BGLU23 as essential components for generating ER-bodies. However, no attempt to determine if co-expression of BGLU21 or BGLU23 and NAI2 or NAI1 forms ER bodies. This point should be included.

Although BGLU21 is the closest homologue of BGLU23, its deficiency caused no significant reduction of the β -glucosidase activity of wild type roots, while deficiency of BGLU23 caused an 80% loss of activity (Plant J., 89: 204-20, 2017). This indicates that the contribution of BGLU21 to total β -glucosidase activity of *A. thaliana* roots is much lower than that of BGLU23. Furthermore, the contribution of BGLU21 to ER-body morphology was also much lower than that of BGLU23 (Plant Cell Physiol., 50: 2015-22, 2009). Additionally, BGLU21 levels are much lower than those of BGLU23 (Plant Cell Physiol., 49: 969-80, 2008). It seemed unlikely that BGLU21 had an important role so we focused on BGLU23.

NAI1 is a transcription factor that regulates the expressions of BGLU23, BGLU21, and NAI2 (Plant Cell, 16: 1536-49, 2004) and is not a homologue of the ER-resident protein NAI2 (Plant Cell, 20: 2529-40, 2008). We thus didn't feel that it would be useful to see if it could substitute for NAI2.

The results also showed that BGLU23 is located in the ER bodies and relocated into the vacuoles in *nai2* mutant. The authors suggest that NAI2 sequestered the BGLU23 to the ER bodies but this is not at all clear and the results supporting the conclusion is weak, given that *nai2* is a ER body-deficient mutant. Whether NAI2 or ER body deficiency leads to the mistargeting of BGLU23 needs to be clarified and sufficiently discussed.

To clarify the point, we did additional experiments and added the results showing the interaction between NAI2 and BGLU23 (Fig. 3, Supplementary Fig. 3, and Supplementary Data 1) the NAI2 requirement for accumulation of BGLU23 and for defense against Woodlice (Fig. 5a and 5c; Supplementary Fig. 5b).

Then, we added the following statement in Discussion:

Although the ER-retention signal KDEL does not perfectly prevent BGLU23 from leaking into the vacuoles, it contributes to the formation of ER bodies. We propose a model of a NAI2-dependent fail-safe defense system, in which BGLU23 proteins are robustly sequestered in ER bodies until they are released by tissue damage. First, the ER-retention signal primarily prevents *de-novo* synthesized BGLU23 proteins from escaping the ER, resulting in increasing the BGLU23 levels in the ER. Second, their high levels enhance the chance of BGLU23 proteins to contact with another ER protein NAI2. Third, NAI2 helps to aggregate BGLU23 proteins in the ER subdomains, giving raise to ER bodies.

The results support that BGLU23 and BGLU21 are major glucosidases generating glucosinolates through metabolome profiling with WT and *bglu23/bglu21* mutant. The effects of *bglu23* and *bglu21* single mutant on metabolome and herbivore damage need to be included in the analysis. Also please explain why not include BGLU21 in the previous experiments.

As stated above (major point 1) BGLU21 seemed to be of minor importance compared to BGLU23 and it would take ~two months and ~\$8,000 to perform the experiments with new mutants. Thus, we would prefer not to do them.

Minor points:

Figure 2c, please add statistical analysis here and indicate how many biological replicates.

To do replicate experiments, we needed multiple transformed cell lines that have stable expression levels of NAI2. But the NAI2 expression levels varied among the lines. Because of the instability, we decided that the rate data was not reliable enough to report and deleted it.

Figure 3, indicate whether they are one- or two-sided Student t-test.

Note that Fig. 3 is now Fig. 1. We indicated that they are two-sided in Fig. 1, Fig. 5, and

Table 1).

Table 1 and Table S1, please add statistical analysis.

We added two-sided Student t-test p values to both tables (Table 1 and Supplementary Table 1).

REVIEWERS' COMMENTS:

Reviewer #1 (Remarks to the Author):

The manuscript is adequately revised.
I have no further comments to make on the current version.

Reviewer #2 (Remarks to the Author):

I have no more concerns.

Reviewer #3 (Remarks to the Author):

The authors performed additional experiments and addressed my questions. I have no more major concerns.

Minor revision:

line 547, is "(ed^(eds))" a typo? Please correct it if it is.

Responses to Reviewers

Reviewer #1 (Remarks to the Author):

The manuscript is adequately revised.

I have no further comments to make on the current version.

Reviewer #2 (Remarks to the Author):

I have no more concerns.

Reviewer #3 (Remarks to the Author):

The authors performed additional experiments and addressed my questions. I have no more major concerns.

Minor revision:

line 547, is "(ed^(eds))" a typo? Please correct it if it is.

Response: Done.